# HI-TPH: A LARGE-SCALE HIERARCHICAL DATASET FOR TCR-PHLA BINDING PREDICTION

## ABSTRACT

The interaction between the T cell receptor (TCR) and peptide-human leukocyte antigen complex (pHLA) is a fundamental process underlying T cell-mediated immunity. Computational methods have been developed to predict TCR-pHLA binding, but most existing models were trained on relatively small datasets and focused solely on the Complementarity Determining Region 3 (CDR3) of the TCR $\beta$ chain. A key barrier to developing advanced prediction models is the limited availability of comprehensive data containing understudied prediction components. In this light, we developed the Hi-TPH dataset with more protein sequences and gene annotations. The dataset is stratified into five hierarchical subsets at four different levels, ranging from Hi-TPH level I with only the peptide sequence and TCR CDR3 $\beta$ to Hi-TPH level II, III, and IV that incorporate increasing levels of HLA sequences, full TCR $\alpha$ and $\beta$ chains, and gene annotations. Hi-TPH at any level represents the largest dataset with corresponding prediction components to date, for instance, the Hi-TPH level IV dataset is at least 5.99 times the size of existing ones regarding the number of TCR-pHLA pairs. We further report benchmark results on the Hi-TPH dataset, establishing valuable baselines for the TCR-pHLA binding prediction task. This comprehensive dataset and associated benchmarks provide a valuable resource for developing advanced TCR-pHLA binding prediction models and exploring research directions such as understanding the contribution of different components and enhancing model generalization to unseen peptides, with potential applications in developing targeted therapies, including personalized vaccines and immunotherapies.

## 1 INTRODUCTION

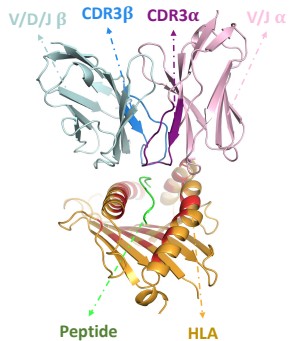
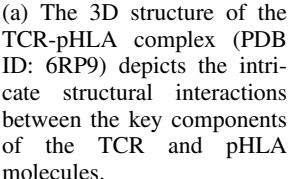
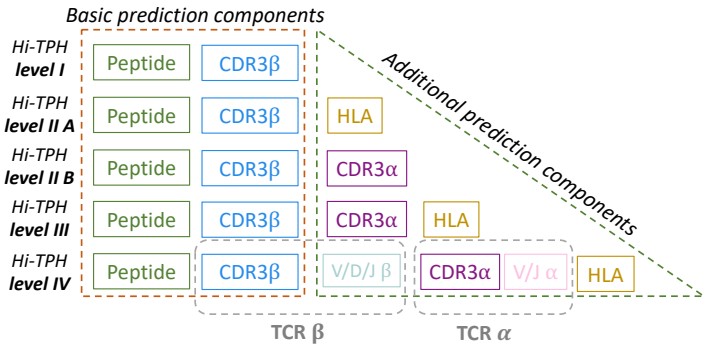

(a) The 3D structure of the TCR-pHLA complex (PDB ID: 6RP9) depicts the intricate structural interactions between the key components of the TCR and pHLA molecules.

(b) The Hi-TPH dataset provides a comprehensive resource for TCR-pHLA binding prediction, encompassing increasing levels of protein sequences and gene annotations of TCR-pHLA binding components. TCR $\alpha$ and TCR $\beta$ denote the full TCR $\alpha$ and TCR $\beta$ chain excluding the constant domain, respectively.

Figure 1: 3D structure illustration of TCR-pHLA interaction and overview of the Hi-TPH dataset.

The interaction of T cell receptor, peptide, and human leukocyte antigen is a crucial process underlying T cell-mediated immunity (Davis & Bjorkman, 1988; Glanville et al., 2017b). This molecular recognition process consists of two key steps: 1) the binding of antigenic peptides and HLA molecules, forming pHLA complexes on the surface of antigen-presenting cells, and 2) the binding of the TCR with the pHLA complex. Identifying the specific TCR-peptide-HLA interactions is of particular significance for the development of effective immunotherapies, as it enables the targeted activation and expansion of T cells capable of recognizing and eliminating infected or cancer cells (Schumacher & Schreiber, 2015; Purcell et al., 2007; Ott et al., 2017).

Computational methods have been developed to predict interactions between TCR, peptide, and HLA molecules (Wang et al., 2023; Hudson et al., 2023; Weber et al., 2024; Meysman et al., 2023). In predicting the binding of peptide and HLA, Mei et al. (Mei et al., 2021) compiled a comprehensive dataset with binding peptide-HLA pairs from multiple sources, which Chu et al. (Chu et al., 2022) then leveraged to develop a transformer-based method that achieved state-of-the-art pHLA binding prediction performance. However, in the task of predicting TCR-pHLA binding, most available computational models are trained on relatively small datasets (Springer et al., 2019; Montemurro et al., 2021; Springer et al., 2021; Xu et al., 2021; Lu et al., 2021; Gao et al., 2023; Peng et al., 2023; Pham et al., 2023). Moreover, the majority of these methods focus solely on the CDR3 domain of the TCR $\beta$ chain. This emphasis is rooted in the understanding that the CDR3 $\beta$ is the most variable segment of the TCR and directly interfaces with the bound peptide within the pHLA complex (Davis & Bjorkman, 1988; Krogsgaard & Davis, 2005), as shown in Fig. 1(a), leading to the hypothesis that it is the primary driving component of T cell specificity (Glanville et al., 2017a).

Emerging evidence from recent studies suggests that other components and regions of TCRs beyond the CDR3 $\beta$ may also contribute significantly to the formation of the TCR-pHLA interaction (Springer et al., 2021; Carter et al., 2019). Specifically, the importance of the HLA molecule, the TCR $\alpha$ chain, and other CDR loops within the TCR $\beta$ chain in mediating immunological recognition process is highlighted (Stadinski et al., 2014; Gruta et al., 2018). Neglecting these additional prediction components may limit the ability of existing methods to capture the intricate sequence patterns from the full TCR chains and other relevant features, thereby constraining their prediction performance (Springer et al., 2021; Fischer et al., 2019). The limited availability of comprehensive data containing understudied TCR and pHLA regions is a key barrier to developing TCR-pHLA binding prediction models that consider these components. Therefore, expanding the scope of collected TCR and pHLA data will be necessary to enable a more complete understanding of the TCR-pHLA interaction.

To address this challenge, we developed the Hi-TPH dataset, which provides data at five hierarchical subsets across four levels. Each higher level contains increasing protein sequences and gene annotations of TCR-pHLA binding components, as shown in Fig. 1(b). Notably, the level IV dataset not only accounts for the TCR variable (V), joining (J), and diversity (D, $\beta$ chain only) gene annotations, but also includes the full TCR $\alpha$ and $\beta$ chain sequences. These full sequence data are derived from complex gene rearrangement and variation processes, providing a rich source of information of TCRs beyond the CDR3s. At any level, Hi-TPH represents the largest dataset to date with the corresponding components for TCR-pHLA binding prediction. For instance, the level IV dataset is at least 5.99 times the size of existing datasets with peptide and full TCR sequences in terms of the number of TCR-pHLA pairs.

To facilitate the development of advanced TCR-pHLA prediction models, we also introduced an on-the-fly mispairing method to generate negative (i.e., non-binding) TCR-pHLA pairs, rather than using a fixed negative dataset as is typical (Gao et al., 2023; Springer et al., 2021). This on-the-fly mispairing approach reduces bias and exposes the model to a wider array of negative examples (Chen et al., 2020), thereby enhancing its ability to distinguish binding TCR-pHLA pairs. Furthermore, we constructed benchmark results on the Hi-TPH dataset using seven different models. These benchmark results provide useful insights in exploring the contribution of additional components and the model generalization to unseen peptides.

The main contributions of this paper are summarized as follows.

- We highlighted the importance of additionally considering the HLA molecule and full sequences of both the TCR $\alpha$ and $\beta$ chains in TCR-pHLA binding prediction models.

- We released Hi-TPH, a large-scale hierarchical dataset for TCR-pHLA binding. Hi-TPH provides five subsets with increasing levels of protein sequences and gene annotations, each with more peptides and TCRs than existing datasets.

- We introduced the on-the-fly mispairing method to dynamically generate non-binding TCR-pHLA pairs for training TCR-pHLA binding prediction models and reported benchmark results with seven different models on the Hi-TPH dataset to facilitate further research.

## 2 RELATED WORK

**Peptide-HLA binding**  Peptide binding to HLA molecules is a critical prerequisite for the formation of TCR-pHLA complex, as it determines the pool of peptides presented to TCRs (Chu et al., 2022; Yewdell & Bennink, 1999). This peptide-HLA binding step involves the anchoring of specific amino acid residues within the peptide sequence to complementary binding pockets in the HLA molecule, resulting in conformational changes that facilitate the display of the pHLA complexes on the surface of antigen-presenting cells. To reduce the cost of wet-lab experiments, numerous computational tools have been developed to predict peptide-HLA binding (Reynisson et al., 2020; Mei et al., 2021; Chu et al., 2022; Zhang et al., 2022; Wang et al., 2023). Notable among these, Mei et al. (Mei et al., 2021) provided a comprehensive dataset containing peptide binders across different HLA class I alleles by collecting data from biological databases, prediction tools, and published studies. Building upon this, Chu et al. (Chu et al., 2022) achieved state-of-the-art pHLA binding prediction results using a transformer-based model. In most current approaches for predicting peptide-HLA binding, the primary inputs used are typically the amino acid sequences of the peptide and the HLA molecule. For basic methods that only consider the protein sequence, they typically only need to take these two elements into account.

**Multiple components for predicting TCR-pHLA binding**  Compared to the relatively straightforward peptide-HLA binding, the binding of TCR-pHLA is a more complex process involving the engagement of multiple molecules and TCR regions. Within TCR variable regions, the CDR1 and CDR2 loops primarily interact with the HLA $\alpha$-helices, while the hypervariable CDR3s predominantly engage with antigenic peptides (Fig. 1(a)) (Davis & Bjorkman, 1988). The CDR3 loops exhibit the highest sequence diversity and serve as the principal determinants of receptor binding specificity, with CDR3 $\beta$ suggested as the main driving component of T cell specificities (Glanville et al., 2017b). While the basic TCR-pHLA binding prediction components are the peptide and TCR CDR3 $\beta$ (Gao et al., 2023; Peng et al., 2023), recent studies have indicated potential contributions from the CDR3 $\alpha$, HLA molecule and CDR1/2 (Carter et al., 2019; Gruta et al., 2018). Advancements in TCR-pHLA binding prediction models, such as NetTCR2.0 (Montemurro et al., 2021), TCRGP (Jokinen et al., 2021), ERGO-II (Springer et al., 2021), and TCRAI (Zhang et al., 2021a), have incorporated the CDR3 $\alpha$ as an additional component. ERGO-II and TCRAI also consider the V, D, and J gene annotations, as well as the HLA sequences, in their models. Furthermore, STA-PLER (Kwee et al., 2023) reconstructs the full TCR $\alpha$ and TCR $\beta$ amino acid sequences from the V(D)J annotations and uses the full sequences to train transformer-based prediction models. While these advancements are promising, the relative contributions of different TCR and pHLA components to the binding process remain incompletely understood, warranting further investigation.

## 3 HI-TPH: HIERARCHICAL TCR-pHLA INTERACTION DATASET

The Hi-TPH dataset is a comprehensive collection of TCR-pHLA binding pairs, encompassing multiple subsets with varying levels of protein sequences and gene annotations. This dataset has been structured into five hierarchical subsets at four different levels, each with a unique set of prediction components, catering to the diverse needs of the research community. Table 1 provides a comparative overview of the Hi-TPH dataset and other publicly available TCR-pHLA datasets. This section describes the process used to generate the Hi-TPH dataset, followed by a detailed characterization of its statistical properties. Additionally, a novel solution utilizing on-the-fly mispairing to generate negative samples is introduced. The Hi-TPH dataset is released under a CC BY-NC 4.0 license.

| | # Samples | | | Protein sequence | | | | | | Gene | | | |
|---|---|---|---|---|---|---|---|---|---|---|---|---|---|
| Dataset | Pairs | Peptides | TCRs | Peptide | CDR3β | HLA | CDR3α | TCRα | TCRβ | V/Jα | V/D/Jβ | HLA type | Data collection |
| PanPep (Gao et al., 2023) | 32,080 | 699 | 29,467 | ✓ | ✓ | | | | | | | HLA I | Curation |
| TEIM (Peng et al., 2023) | 45,481 | 355 | 44,227 | ✓ | ✓ | | | | | | | HLA I | Curation |
| DLpTCR (Xu et al., 2021) | 7,121 | 304 | 6,583 | ✓ | ✓ | | | | | | | HLA I | Curation |
| NetTCR-2.0 (Montemurro et al., 2021) | 11,431 | 17 | 11,425 | ✓ | ✓ | | ✓* | | | | | HLA I | Curation |
| epiTCR (Pham et al., 2023) | 66,471 | 1,391 | 61,159 | ✓ | ✓ | ✓ | | | | | | HLA I | Curation |
| pMTnet (Lu et al., 2021) | 32,070 | 602 | 28,864 | ✓ | ✓ | ✓ | | | | | | HLA I | Curation |
| TCRAI (Zhang et al., 2021a) | 8,130 | 16 | 8,101 | ✓ | ✓ | ✓ | ✓ | | | ✓* | ✓* | HLA I/II | Curation + Experiment |
| ERGO-II (Springer et al., 2021) | 27,260 | 389 | 26,548 | ✓ | ✓ | ✓ | ✓* | | | ✓* | ✓* | HLA I/II | Curation |
| STAPLER (Kwee et al., 2023) | 4,457 | 604 | 4,253 | ✓ | ✓ | ✓ | ✓ | ✓ | ✓ | ✓ | ✓ | HLA I | Curation |
| Hi-TPH-level I | 214,641 | 1,647 | 204,376 | ✓ | ✓ | | | | | | | | |
| Hi-TPH-level II A | 78,679 | 1,401 | 73,553 | ✓ | ✓ | ✓ | | | | | | | |
| Hi-TPH-level II B | 28,375 | 1,154 | 26,502 | ✓ | ✓ | | ✓ | | | | | HLA I | Curation |
| Hi-TPH-level III | 28,262 | 1,148 | 26,353 | ✓ | ✓ | ✓ | ✓ | | | | | | |
| Hi-TPH-level IV | 26,704 | 927 | 24,639 | ✓ | ✓ | ✓ | ✓ | ✓ | ✓ | ✓ | ✓ | | |

Table 1: A comparative overview of the characteristics of publicly available datasets from published studies and the Hi-TPH dataset. The statistical information presented here was obtained in April 2024, and only the positive data (i.e., binding TCR-pHLA pair) is considered. The symbol ✓* indicates that the corresponding information is partially available in the respective dataset. The TCR $\alpha$ and $\beta$ sequence fields in the table refer to the full-length TCR $\alpha$ and $\beta$ chains, including the variable (V), diversity (D), and joining (J) regions, excluding the constant (C) domains.

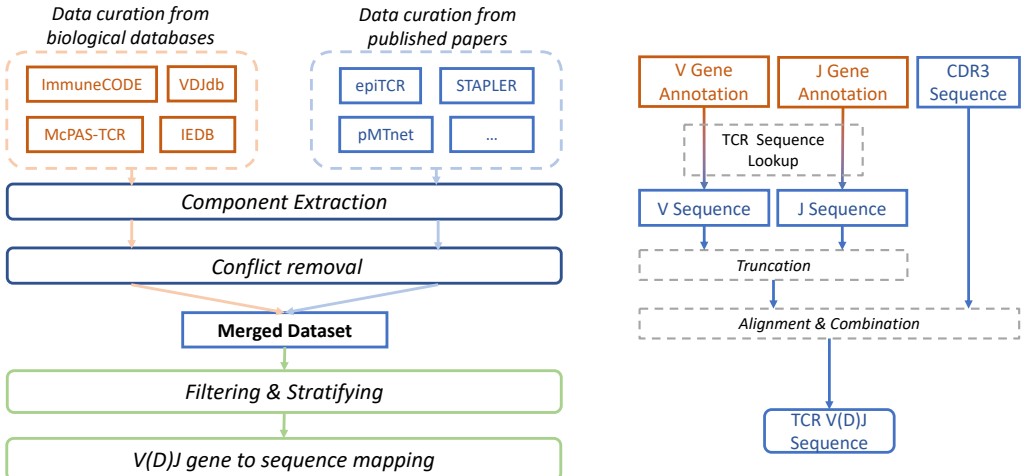

(a) Positive data collection: The workflow begins with data curation from various biological databases such as ImmuneCODE, VDJdb, McPAS-TCR, and IEDB, alongside published papers like epiTCR, STAPLER, and pMTnet. The initial stages involve component extraction followed by the resolution of conflicts between datasets, leading to a merged dataset. Subsequent steps include filtering and stratifying the data, culminating in the mapping of V(D)J genes to their corresponding sequences.

(b) V(D)J gene to sequence mapping: The workflow for mapping V(D)J genes to sequences begins with the annotation of V and J genes, followed by a TCR sequence lookup and the extraction of the CDR3 sequence. Subsequent truncation and alignment steps culminate in the generation of the final TCR V(D)J sequence.

Figure 2: The process of data collection for Hi-TPH positive data.

### 3.1 DATASET GENERATION

#### 3.1.1 POSITIVE DATA COLLECTION

Building upon the wealth of publicly available biological databases and published prediction models, we have collected and curated a comprehensive dataset of binding TCR-pHLA pairs, as depicted in Fig. 2(a). Specifically, we accessed data from Immune Epitope Database (IEDB) (Vita et al., 2018), McPAS-TCR (Tickotsky et al., 2017), the VDJdb (Bagaev et al., 2019) and ImmuneCODE-MIRA (Nolan et al., 2020), all of which serve as authoritative repositories of TCR-pHLA binding information. The IEDB represents a meticulously curated and regularly updated repository, housing positive TCR-peptide pairs originating from both human and non-human sources. The McPAS-TCR database serves as a comprehensive compendium of disease-associated positive TCR-peptide pairs, drawn from both human and murine sources. VDJdb stands as a manually curated and regularly updated database encompassing positive TCR-peptide pairs across mouse and human origins. ImmuneCODE-MIRA maps TCRs binding to SARS-Cov-2 virus epitopes. Furthermore, we included TCR-pHLA binding pairs reported in published papers (Pham et al., 2023; Kwee et al., 2023; Lu et al., 2021), further enriching our dataset. For each data resource, we extract prediction components including two main parts: 1) protein sequences, including peptide, CDR3 $\beta$, CDR3 $\alpha$, HLA and the full TCR $\alpha$ and TCR $\beta$ chains; 2) gene annotations, including V/J $\alpha$ genes and V/D/J $\beta$ genes. Conflict removal is performed to ensure data quality. For ImmuneCODE-MIRA, we filtered out non-unique peptide-TCR samples. By focusing solely on positive, experimentally validated binding TCR-pHLA pairs, we have assembled a robust dataset that enables the rigorous study of TCR-pHLA binding prediction and its underlying mechanisms.

#### 3.1.2 FILTERING AND STRATIFYING

Recent studies suggest that, in addition to CDR3 $\beta$, other regions of the TCR can also make substantial contributions to the formation of the TCR-pHLA complex (Springer et al., 2021; Carter et al., 2019; Stadinski et al., 2014; Gruta et al., 2018). This expanding understanding highlights the need for comprehensive examination of the various components involved in TCR-pHLA binding. Therefore, we stratified the dataset into five subsets at four different levels with increasing information of prediction components. At the most basic level I, the data includes only the CDR3 $\beta$ paired with the peptide sequence. Level II A incorporates the HLA allele information and the corresponding protein sequence, while level II B contains the CDR3 sequence of the TCR $\alpha$ chain. Advancing to level III, the dataset combines CDR3 domains of both TCR $\alpha$ and $\beta$ chains, along with the peptide and HLA sequences. Reaching the most detailed level of the Hi-TPH dataset, level IV, encompasses the full TCR $\alpha$ and $\beta$ chains, in addition to the peptide and HLA sequences. In this comprehensive level, only the constant domains of the TCRs are excluded. By organizing the Hi-TPH dataset in this hierarchical manner, we aim to empower researchers to systematically investigate the relative contributions of diverse components in shaping TCR-pHLA binding interactions and, ultimately, T cell activation and antigen recognition.

#### 3.1.3 GENE TO SEQUENCE MAPPING OF TCR V/J $\alpha$ AND V/D/J $\beta$

As prior research has demonstrated, CDR1, CDR2, and other TCR $\alpha$ and TCR $\beta$ regions can contribute to TCR-pHLA binding (Stadinski et al., 2014; Harris et al., 2016). While our level IV dataset includes V/J gene annotations for TCR $\alpha$ and V/D/J gene annotations for TCR $\beta$, the full TCR $\alpha$ and TCR $\beta$ chain sequences are only partially available. To address this, we performed a gene-to-sequence mapping procedure to generate the full TCR $\alpha$ and TCR $\beta$ chains, thereby achieving gene-sequence alignment for the Hi-TPH-level IV dataset. Following the method in previous studies (Kwee et al., 2023), we reconstructed the full TCR $\alpha$ and TCR $\beta$ sequences (Fig. 2(b)). From the V and J gene annotations, we conducted sequence lookup based on the IMGT/Gene-DB (Giudicelli et al., 2005). The sequences were then truncated and combined with the CDR3 $\alpha$ or CDR3 $\beta$ to form the full TCR $\alpha$ or TCR $\beta$ sequences, excluding the constant domains.

### 3.2 DATASET STATISTICS

**Dataset size magnification** The number of binding TCR-pHLA pairs, TCRs and peptides contained in different levels of the Hi-TPH dataset are illustrated in Table 1. In Fig. 3, we demonstrate

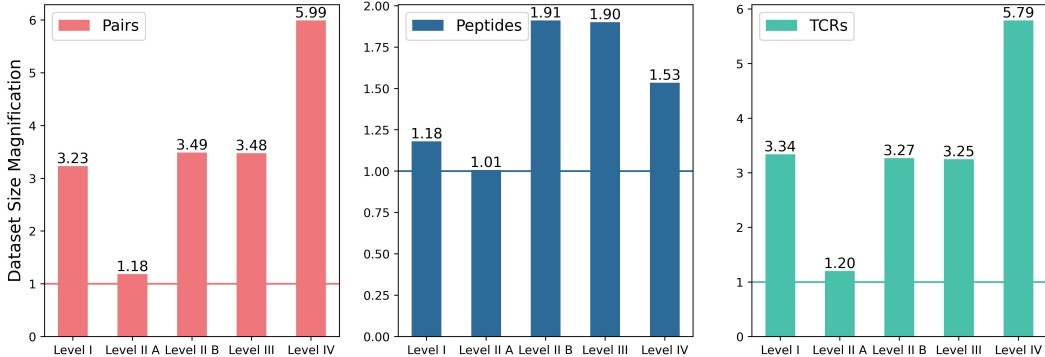

Figure 3: Dataset size magnification. The $y$-axis shows the magnification factor compared to the largest existing dataset with the same prediction components, for datasets at different Hi-TPH levels. The magnification factor is displayed based on the number of TCR-pHLA pairs, peptides, and TCRs, in the left, middle, and right panels, respectively.

the magnification of Hi-TPH compared to the current largest dataset containing the same level of prediction components. It is evident that the Hi-TPH dataset includes more records than the largest existing dataset, in terms of binding pairs, peptides, or TCRs. For example, the level I dataset is approximately 3.23 times the size of the existing dataset, containing around 215,000 binding TCR-pHLA pairs. Furthermore, the level IV dataset, which includes the most comprehensive prediction components from protein sequences to gene annotations, is 5.99 times the size of the previously largest dataset of comparable scope, providing a significantly expanded resource for research.

**Long-tail distribution of binding TCRs for different peptides**   As shown in Fig. 4(a), the number of binding TCRs varies across different peptides, following a typical long-tail distribution. The peptides located in the head exhibit numerous binding TCRs, numbering in the hundreds or thousands, while the majority of peptides are located in the tail, with only a small fraction of binders. This long-tail pattern suggests that models need to be carefully designed to address the bias towards the head peptides and overcome the significant data sparsity issues for the tail peptides (Gao et al., 2023; Zhang et al., 2021b).

**Length distribution of peptide and CDR3 sequence**   Fig. 4(b) summarizes the distribution of peptide lengths at different levels in the Hi-TPH dataset, excluding peptide sequences with outlier lengths (i.e., more than 25 amino acids). The distribution is centered around 9-mer peptides, consistent with features of HLA-presented peptides in existing datasets (Mei et al., 2021; Reynisson et al., 2020). Similarly, Fig. 4(c) illustrates the comparable length distribution of paired CDR3 $\alpha$ and CDR3 $\beta$ sequences in the level III dataset, with the majority falling within the 10-20 amino acid range and peaking around 15 residues. This characteristic bias towards 9-mer peptides and 15-mer CDR3s should be considered during model development. Models trained on this dataset may inherently exhibit similar biases, potentially leading to suboptimal performance on sequences of other lengths (Chu et al., 2022; Peng et al., 2023). Strategies such as data augmentation (Sun et al., 2024), transfer learning (Gao et al., 2023), or architectural modifications (Gao et al., 2023) may be required to improve model generalization across a wider range of peptide and CDR3 lengths.

### 3.3 On-the-Fly Mispairing for Negative Data Generation

The variable region of TCR chains is formed by the assembly of V, D (for $\beta$ chains), and J gene segments, with additional nucleotide additions and deletions at the junctional regions further expanding TCR diversity (Mora & Walczak, 2019). The diversity of pHLA complexes is facilitated by the peptide-binding specificities of HLA molecules. Given the vast TCR and pHLA sequence space, the typical approach to generate negative TCR-pHLA pairs is to randomly pair a TCR from the repertoire with a peptide (Springer et al., 2021; Lu et al., 2021; Kwee et al., 2023; Zhang et al., 2021a), resulting in a low probability of generating false negatives.

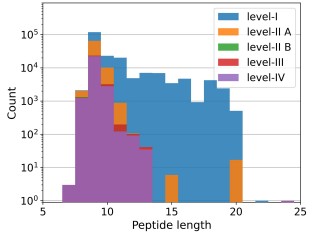 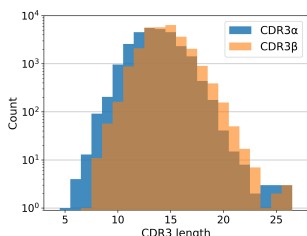

(a) Long-tail distribution of the number of binding TCRs for different peptides in different levels of Hi-TPH.

(b) Peptide length distribution in different levels of Hi-TPH.

(c) CDR3 length distribution in Hi-TPH-level III.

Figure 4: Statistical analysis of subsets at different levels of Hi-TPH.

However, the use of fixed negative data has notable limitations. The predetermined set of non-binding TCR-pHLA pairs does not accurately reflect the true distribution of such complexes, leading to biases in the training and evaluation processes (Chen et al., 2020; Dens et al., 2023; Moris et al., 2020). Rather than fixed negatives, we opt for dynamic sampling during training (Yang et al., 2024). For each positive pair, we randomly select a non-binding TCR and pair it with the peptide to generate a negative sample. Crucially, the negative samples differ in each training epoch, leveraging the vast TCR and pHLA repertoires to efficiently generate non-binding pairs. This on-the-fly mispairing approach would reduce bias and exposes the model to a wider array of negative examples, enhancing its ability to distinguish binding TCR-pHLA pairs (Chen et al., 2020; Yang et al., 2024). We experimentally validated the efficacy of the proposed on-the-fly mispairing technique. The details are listed in Appendix A.5.

## 4 BENCHMARKS

### 4.1 EXPERIMENT SETUP

To demonstrate the usage of the Hi-TPH dataset and provide comprehensive benchmark results, we followed previous studies on TCR-pHLA binding prediction (Mora & Walczak, 2019; Xu et al., 2021; Gao et al., 2023; Springer et al., 2021; Lu et al., 2021; Kwee et al., 2023; Montemurro et al., 2021; Zhang et al., 2021a; Meysman et al., 2023; Weber et al., 2024) and adopted a binary classification task to predict whether a given TCR binds to the pHLA complex. The dataset at each level was randomly split into 8:1:1 training, validation, and test sets. We evaluated different models that take various types of protein sequence information as inputs, corresponding to the different levels of the Hi-TPH dataset hierarchy: level I (peptide, CDR3 $\beta$), level II A (peptide, HLA, CDR3 $\beta$), level II B (peptide, CDR3 $\alpha$, CDR3 $\beta$), level III (peptide, HLA, CDR3 $\alpha$, CDR3 $\beta$), and level IV (TCR $\alpha$, TCR $\beta$, peptide, HLA). This approach enabled the evaluation of model performance with increasing information depth from the Hi-TPH dataset.

During training, the on-the-fly mispairing approach was employed to dynamically generate non-binding TCR-pHLA pairs. The evaluation involved binary classification on the test set, with negative samples randomly generated five times, and the average AUC, accuracy, and F1-score were calculated across these five runs. Further experimental details are provided in Appendix A.3.

### 4.2 EVALUATED MODELS

The benchmark evaluated the following models in three different categories:

- **Traditional machine learning models**, including the Random Forest (**RF**) algorithm employed in epiTCR (Pham et al., 2023).

- **Simple neural network-based models**, comprising a Multi-Layer Perceptron (**MLP**) and a Long Short-Term Memory (**LSTM**) network, as utilized in pMTnet (Lu et al., 2021), TCRAI (Zhang et al., 2021a), and ERGO-II (Springer et al., 2021).

| Model | Level I | | | Level II A | | | Level II B | | | Level III | | | Level IV | | |
|---|---|---|---|---|---|---|---|---|---|---|---|---|---|---|---|
| | AUC | F1 | ACC | AUC | F1 | ACC | AUC | F1 | ACC | AUC | F1 | ACC | AUC | F1 | ACC |
| RF | 0.7322 | 0.6802 | 0.6551 | 0.7252 | 0.6757 | 0.6536 | 0.6026 | 0.5986 | 0.5651 | 0.5963 | 0.5669 | 0.5542 | 0.6375 | 0.6195 | 0.5895 |
| MLP | 0.7561 | 0.6550 | 0.6709 | 0.7313 | 0.6388 | 0.6513 | 0.6171 | 0.5308 | 0.5587 | 0.6163 | 0.5369 | 0.5689 | 0.6320 | 0.5605 | 0.5792 |
| LSTM | 0.7463 | 0.7004 | 0.6695 | 0.7360 | **0.7140** | 0.6631 | 0.6221 | **0.6375** | 0.5777 | 0.6065 | 0.6060 | 0.5708 | 0.5885 | 0.6420 | 0.5598 |
| ESM2-8M | **0.7893** | 0.6741 | 0.6950 | 0.7506 | 0.6753 | 0.6702 | 0.6332 | 0.5586 | 0.5791 | 0.6313 | **0.6343** | 0.5777 | **0.6531** | 0.5829 | 0.5904 |
| ESM2-35M | 0.7822 | **0.7070** | **0.6972** | **0.7548** | 0.7094 | **0.6794** | **0.6404** | 0.5810 | **0.5887** | 0.6363 | 0.5962 | **0.5913** | 0.6510 | **0.6526** | **0.6044** |
| ESM2-150M | 0.7876 | 0.6659 | 0.6944 | 0.7541 | 0.6996 | 0.6771 | 0.6178 | 0.5821 | 0.5703 | **0.6492** | 0.6312 | 0.5911 | 0.6222 | 0.6454 | 0.5904 |
| TAPE-BERT | 0.7740 | 0.6873 | 0.6886 | 0.7526 | 0.6901 | 0.6728 | 0.6300 | 0.5878 | 0.5825 | 0.6283 | 0.5995 | 0.5822 | 0.6308 | 0.5894 | 0.5823 |

Table 2: Benchmark results of different models on datasets at different levels of Hi-TPH. The best results for each metric are shown in bold. The final AUC, accuracy (ACC), and F1-score (F1) are calculated as the mean across 5 runs with randomly generated negative samples.

- **Protein language model (PLM)-based models** (Hu et al., 2022), encompassing **TAPE-BERT** (Rao et al., 2019), a BERT-base (Devlin et al., 2019) model pre-trained on Pfam (El-Gebali et al., 2018) with 92M parameters, and three variants of the ESM2 family (Lin et al., 2022), denoted as **ESM2-8M**, **ESM2-35M**, and **ESM2-150M**, all pre-trained on UniRef50 (Suzek et al., 2014). A basic MLP was added as a projection head and trained alongside the PLM parameters using a supervised fine-tuning approach.

## 4.3 RESULTS

**Main results**   The benchmark results are summarized in Table 2. Key observations are as follows:

- The PLM-based methods outperform RF, MLP, and LSTM in most cases. This superior performance can be attributed to the PLMs' pre-training on large-scale protein sequence datasets. By leveraging universal protein knowledge, the PLMs can capture more generalizable and informative features compared to models trained solely on task-specific data.

- Most models encounter a performance drop when using a higher-level dataset compared to the level I dataset, possibly due to the level I dataset having at least 2 times more binding pairs than the higher-level datasets. The scale of the dataset is critical for predicting TCR-pHLA binding. For this task, methods are needed that can leverage large-scale datasets containing only basic peptide and CDR3 $\beta$ sequences, as well as relatively smaller datasets with additional components like HLA and other TCR variable regions.

- For the PLM-based methods, continuously increasing the number of model parameters does not necessarily lead to improved performance, especially given the limited data availability. For example, ESM2-35M achieves higher performance than ESM2-150M in most cases. This observation suggests that the larger model, ESM2-150M, may be prone to overfitting on the limited training data, despite its higher parameter count.

**Contribution of additional components**   We further investigate contributions of additional prediction components in each subset of Hi-TPH. We select ESM2-35M and compare its performance when trained solely on peptide and CDR3 $\beta$ sequences, versus when trained on all available components. The results are summarized in Fig. 5(a). The contribution of the HLA component appears to be limited, as the model trained using only the peptide and CDR3 $\beta$ has comparable performance with the model trained using additional HLA sequence. This finding suggests that the peptide and CDR3 $\beta$ features may be the primary drivers of TCR-pHLA binding prediction, and the HLA sequence may not provide significant additional predictive capability. In contrast, the importance of the CDR3 $\alpha$ is particularly evident, leading to more than 15% relative improvement in performance. This suggests that the CDR3 $\alpha$ plays a crucial role in TCR-pHLA binding prediction, and its inclusion in the model significantly enhances the predictive capability. Further, incorporating other variable TCR regions, beyond just the CDR3, provides additional gains. The largest relative improvement is observed on the Hi-TPH-level IV dataset, highlighting the value of considering the full TCR sequences.

**Generalization to unseen peptides**   The ability of TCR-pHLA prediction models to generalize to new peptides remains a significant challenge in this field (Weber et al., 2024). To address this issue, we conducted experiments to evaluate the performance of various baseline models on both seen and unseen peptides. The results, summarized in Table 3, reveal a noticeable drop in performance

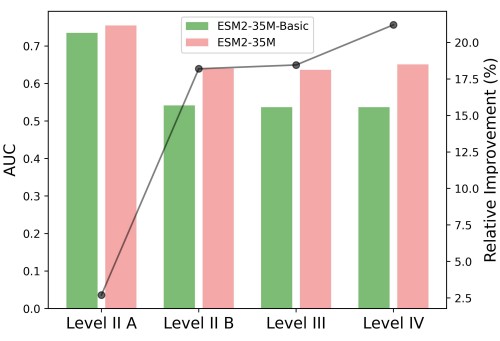 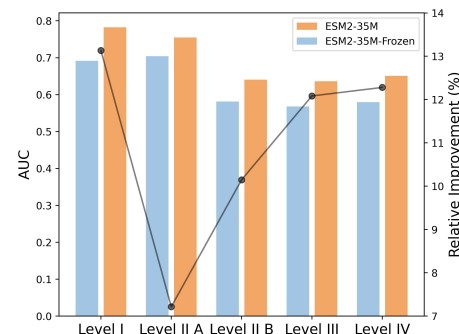

(a) Performance of ESM2-35M on Hi-TPH datasets trained using only the peptide and CDR3 $\beta$ sequences (ESM2-35M-Basic) and all available components (ESM2-35M).

(b) Performance of ESM2-35M on Hi-TPH datasets. Two variants are evaluated: one with the PLM's parameters frozen (ESM2-35M-Frozen) and the other fine-tuned (ESM2-35M).

Figure 5: The absolute AUC performance (bar chart) and the relative improvement (line plot) of the ESM2-35M model.

| Model | Level I | | | Level II A | | | Level II B | | | Level III | | | Level IV | | |
|---|---|---|---|---|---|---|---|---|---|---|---|---|---|---|---|
| | seen | unseen | RD (%) | seen | unseen | RD (%) | seen | unseen | RD (%) | seen | unseen | RD (%) | seen | unseen | RD (%) |
| RF | 0.7333 | 0.6927 | **5.53** | 0.7235 | 0.7067 | 2.32 | 0.6005 | 0.5964 | 0.68 | 0.5934 | 0.6060 | -2.12 | 0.6382 | 0.5402 | 15.35 |
| MLP | 0.7559 | 0.6593 | 12.78 | 0.7324 | 0.6899 | 5.80 | 0.6138 | 0.5821 | 5.16 | 0.6079 | 0.5177 | 14.84 | 0.6318 | 0.5809 | 8.06 |
| LSTM | 0.7473 | 0.6426 | 14.01 | 0.7374 | 0.7096 | 3.77 | 0.6159 | 0.6109 | 0.81 | 0.5983 | 0.5267 | 11.97 | 0.5904 | 0.4929 | 16.51 |
| ESM2-8M | 0.7909 | 0.7129 | 9.86 | 0.7499 | 0.7209 | 3.87 | 0.6278 | 0.5877 | 6.39 | 0.6234 | 0.6470 | **-3.79** | 0.6559 | 0.5834 | 11.06 |
| ESM2-35M | 0.7841 | 0.7050 | 10.09 | 0.7529 | 0.7270 | 3.44 | 0.6349 | 0.6371 | -0.35 | 0.6297 | 0.6363 | -1.05 | 0.6474 | 0.5696 | 12.02 |
| ESM2-150M | 0.7900 | 0.6980 | 11.65 | 0.7535 | 0.7379 | **2.07** | 0.6163 | 0.6522 | **-5.84** | 0.6399 | 0.6208 | 2.98 | 0.6248 | 0.6346 | **-1.57** |
| TAPE-BERT | 0.7756 | 0.7003 | 9.71 | 0.7528 | 0.7161 | 4.88 | 0.6272 | 0.5954 | 5.07 | 0.6267 | 0.5849 | 6.67 | 0.6365 | 0.5686 | 10.67 |

Table 3: Performance comparison of baseline models on seen peptides and unseen peptides. RD denotes relative drop of performance on unseen peptides compared to seen peptides. Results indicating the lowest performance drop are highlighted in bold.

when predicting binding to unseen peptides for most models, even with an expanded dataset. In terms of absolute performance, PLM-based methods outperform others on unseen peptides. While most methods experience a decline when comparing unseen peptides to seen peptides, PLM-based methods show a relatively smaller drop, with some even achieving comparable performance. For instance, ESM2-150M encounters the lowest performance drop at level II A, II B, and IV. This superiority can be attributed to their enhanced representation of protein sequences, underscoring the potential of PLM approaches in advancing TCR-pHLA binding predictions.

**Impact of fine-tuning PLMs** We also explore the impact of fine-tuning PLMs. ESM2-35M-Frozen uses the PLM as a fixed feature extractor, with only the projection head trained, while ESM2-35M fine-tunes the entire PLM. As shown in Fig. 5(b), ESM2-35M outperforms ESM2-35M-Frozen across all datasets. Notably, the most significant improvement was observed in the level I dataset, suggesting that the fine-tuning process is especially advantageous for large-scale TCR-pHLA sequences. Furthermore, the relative improvement also exceeds 10% in the level II B, III, and IV datasets, which can be attributed to the model's enhanced ability to capture the intricacies of more protein sequences when fine-tuned, as opposed to using a frozen PLM. This suggests the fine-tuning process enhances the model's adaptability to specific datasets, allowing it to leverage contextual information that is critical for accurately predicting TCR-pHLA binding. In contrast, the frozen feature extractor is less effective at leveraging the domain-specific information.

## 5 DISCUSSION

**Protein sequence of HLA molecules** The HLA molecules are comprised of $\alpha$ and $\beta$ chains. Existing works typically select the positions that directly interact with the binding peptides (i.e., the contact positions) to represent the HLA molecule (Reynisson et al., 2020; Chu et al., 2022). However, in TCR-pHLA binding, the HLA molecules are hypothesized to interact with the CDR1 and CDR2 loops in the TCR $\alpha$ and $\beta$ chains (Carter et al., 2019). Recent studies also observe

the interaction of HLA with the CDR3 (Gruta et al., 2018). Therefore, using solely the contact positions to represent the HLA molecules will likely not capture the full complexity of the TCR-pHLA interactions. To address this issue, we collect the full HLA protein sequence and the $\alpha 1$ and $\alpha 2$ domains of HLA molecules (i.e., the clipped sequence) for the further exploration of the contribution of HLA to TCR-pHLA binding.

**Limitations and future works**  The primary limitation arises from the binary nature of the binding labels, which only denote whether a TCR-pHLA interaction is considered binding or non-binding. The dataset lacks a continuous confidence score that would differentiate between strongly binding and weakly binding interactions. The absence of such nuanced binding information may constrain the ability of models trained on Hi-TPH to capture the full range of binding affinities. To expand the utility of the Hi-TPH dataset, we plan to integrate it with additional databases in the future. Furthermore, we will collect and release datasets containing 3D structural information for TCR and pHLA complexes, which could facilitate the development of models capable of predicting TCR-pHLA binding or docking using structural inputs (Bradley, 2022; Yin et al., 2023). Additionally, we aim to complete the V(D)J gene to sequence mapping for rare genes, enhancing the dataset's representation of the TCR repertoire.

## 6  CONCLUSION

The TCR-pHLA interaction is a fundamental process underlying T cell-mediated immunity, with critical implications for the development of immunotherapies. While computational methods have emerged to predict TCR-pHLA binding, this task has traditionally been limited by the availability of comprehensive datasets considering multiple prediction components. In this work, we addressed this challenge by introducing the Hi-TPH dataset. By stratifying the data into multiple levels, Hi-TPH provides protein sequences and gene annotations at different levels of granularity for TCR-pHLA prediction, including the HLA molecule and full sequences of both the TCR $\alpha$ and $\beta$ chains. This hierarchical structure enables more detailed and nuanced modeling of the complex TCR-pHLA interaction. At each level, Hi-TPH represents the largest binding TCR-pHLA dataset to date. To aid the development of advanced prediction models, we introduced an on-the-fly mispairing method to dynamically generate non-binding TCR-pHLA pairs during training. We further reported benchmark results with seven models on the Hi-TPH dataset, ranging from traditional machine learning models to PLM-based models. The release of the Hi-TPH dataset, along with the accompanying benchmark results, offers a valuable resource for TCR-pHLA binding prediction research.

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

## A  APPENDIX

### A.1  ACCESS INSTRUCTION

#### A.1.1  DATA FORMAT

The data files within the Hi-TPH dataset are stored in the standard CSV format. This facilitates easy reading, editing, and utilization of the data using a wide range of data processing tools and programming languages.

#### A.1.2  LICENSE

We, the authors, take full responsibility for the content of this dataset and ensure that no rights have been violated in its creation or publication. The Hi-TPH dataset is licensed under the Creative Commons Attribution-NonCommercial 4.0 International (CC BY-NC 4.0) license, meaning it is freely available for non-commercial use, sharing, and adaptation with proper attribution.

#### A.1.3  HOSTING PLAN

The Hi-TPH dataset will continue to be hosted on the GitHub platform for the foreseeable future, and any updates will also be released through the same channel. Furthermore, the dataset is expected to be updated at least one more time within the next year.

### A.2  DATA ACQUISITION

#### A.2.1  INCLUSION CRITERIA FOR BIOLOGICAL DATABASES

The majority of the data in our dataset were from four commonly used databases: IEDB Vita et al. (2018), VDJdb Bagaev et al. (2019), McPAS-TCR Tickotsky et al. (2017), and ImmuneCODE-MIRA Nolan et al. (2020). We accessed the most up-to-date versions of these databases in April 2024 through their online websites[1][2][3][4]. TCR-pHLA pairs from IEDB were retrieved by setting three filters (Epitope: "Linear Sequence", Assay: "T cell (Positive)" and Host: "Human") and pressing "Export Results" for "Receptors". The latest complete database of the other three were downloaded: VDJdb (2023-06-01 version), McPAS-TCR (2022-9-10 version), ImmuneCODE-MIRA (release-002.1 version). We only retained the HLA class I related records for subsequent data preprocessing. In addition, we obtained reference amino acids sequences of HLA alleles[5] and Human TCR genes[6] in FASTA format from IMGT Lefranc (2011) for sequence lookup.

When processing ImmuneCODE-MIRA data, we implemented a rigorous filtering process to exclude samples where the TCR exhibited binding to a group of peptides rather than to a specific peptide. This step was crucial in mitigating potential data errors and ensuring the integrity of our dataset, thereby enhancing the reliability of subsequent analyses.

#### A.2.2  DATA COLLECTION FROM PUBLISH PAPERS

We also collected publicly shared data from nine published papers: PanPep[7] Gao et al. (2023), TEIM[8] Peng et al. (2023), DLpTCR[9] Xu et al. (2021), NetTCR-2.0[10] Montemurro et al. (2021),

---

[1]https://www.iedb.org
[2]https://github.com/antigenomics/vdjdb-db/releases/tag/2023-06-01
[3]http://friedmanlab.weizmann.ac.il/McPAS-TCR
[4]https://clients.adaptivebiotech.com/pub/covid-2020
[5]https://github.com/ANHIG/IMGTHLA/blob/Latest
[6]https://www.imgt.org/vquest/refseqh.html
[7]https://github.com/bm2-lab/PanPep/tree/main/Data
[8]https://github.com/pengxingang/TEIM/tree/main/data/binding_data
[9]http://jianglab.org.cn/DLpTCR/Download
[10]https://github.com/mnielLab/NetTCR-2.0/tree/main/data

| Dataset | # Samples | | | |
| --- | --- | --- | --- | --- |
| | Training | Validation | Test | Total |
| level I | 148,122 | 18,515 | 18,516 | 185,153 |
| level II A | 61,236 | 7,654 | 7,655 | 76,545 |
| level II B | 22,088 | 2,761 | 2,762 | 27,611 |
| level III | 21,982 | 2,747 | 2,749 | 27,478 |
| level IV | 20,789 | 2,598 | 2,600 | 25,987 |

Table 4: Number of positive samples used for benchmarks.

epiTCR[11] Pham et al. (2023), pMTnet[12] Lu et al. (2021), TCRAI[13] Zhang et al. (2021a), ERGO-II[14] Springer et al. (2021), and STAPLER[15] Kwee et al. (2023). We focused on the main data used in each paper while excluding external data with minimal significance, and then conducted statistical analysis and processing on these primary data. Furthermore, most of these public data are accessible under permissive licenses, such as the MIT license, GPL license, or Non-Commercial Use licenses, along with their respective codes.

### A.3 BENCHMARK SETUP

#### A.3.1 THE HI-TPH DATASET IN BENCHMARK

When constructing the benchmark data, we filtered the peptide, CDR3, and full TCR $\alpha$ and $\beta$ sequences in the Hi-TPH dataset by sequence length. Drawing on previous studies Chu et al. (2022); Peng et al. (2023) and length distribution statistics, we limited the peptide length to 8-15 for level I and 8-10 for levels II-IV, the CDR3 length to 9-19 for levels I-IV, and the full TCR $\alpha$ and $\beta$ sequence lengths to 105-121 and 109-121, respectively, for level IV. These length filters were applied to ensure the sequences were well-suited for the padding operations required during model training. The goal was to create a robust benchmark dataset, avoiding potential issues caused by extreme sequence lengths or outliers. The filtered TCR-pHLA pairs were then used for training/validation/test set division as well as negative data generation. Overall, while using the Hi-TPH dataset as the basis for our experiments, we only made such targeted adjustments to a small number of individual data points. The statistics of the number of positive samples for the benchmarks are shown in Table 4.

#### A.3.2 MODEL IMPLEMENTATIONS

Seven models were utilized in our benchmark, including Random Forest (**RF**), Multi-Layer Perceptron (**MLP**), Long Short-Term Memory (**LSTM**), and four Protein Language Model (PLM)-based models. The amino acid sequences of pHLA and TCR were padded with the special character "X" (denoting unknown or arbitrary amino acids) to ensure equal length, and the concatenated TCR-pHLA sequences were then fed into each model. The model implementations are as follows:

- **RF**. We implemented RF using scikit-learn Pedregosa et al. (2011). One-hot vectors of TCR-pHLA sequences were flattened as inputs to RF.

- **MLP**. We implemented a basic MLP in Pytorch Paszke et al. (2019), using one embedding layer and two hidden layers with ReLU activation. The embedding layer converts amino acids in the TCR-pHLA sequence into vectors, which are then flattened and fed into the subsequent hidden layers.

- **LSTM**. We also implemented an LSTM recurrent neural network using Pytorch, comprising an embedding layer as in MLP, an LSTM layer and a fully connected layer for classification. The embedding layer converts the amino acids in the TCR-pHLA sequence into vectors for input into

---

[11]https://github.com/ddiem-ri-4D/epiTCR/tree/main/data/finalData

[12]https://github.com/tianshilu/pMTnet/tree/master/data

[13]https://github.com/regeneron-mpds/TCRAI/tree/main/data

[14]https://github.com/IdoSpringer/ERGO-II/tree/master/Samples

[15]https://files.aiforoncology.nl/stapler/data/

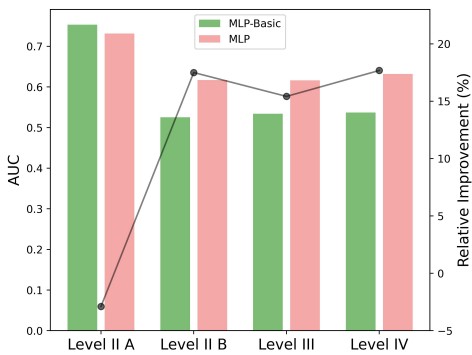 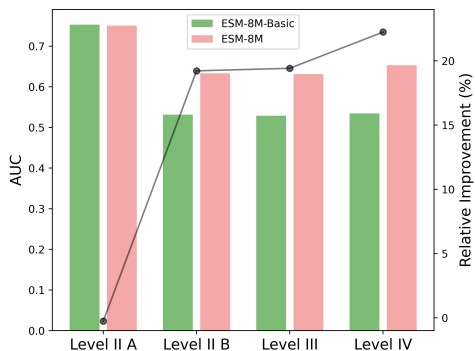

Figure 6: Performance of MLP on Hi-TPH datasets trained using only the peptide and CDR3 $\beta$ sequences (MLP-Basic) and all available components (MLP).

Figure 7: Performance of ESM2-8M on Hi-TPH datasets trained using only the peptide and CDR3 $\beta$ sequences (ESM2-8M-Basic) and all available components (ESM2-8M).

the LSTM layer, and the average output from the LSTM layer is used by the final fully connected layer.

- **PLM-based models**. We added the MLP on pre-trained PLMs to customize them for the classification task. The included PLMs are ESM2-8M, ESM2-35M, ESM2-150M Lin et al. (2022), and TAPE-BERT Rao et al. (2019). The pre-trained ESM-2-8M/35M/150M models have been released on HuggingFace[16][17][18], whereas the pre-trained TAPE-BERT model is based on the original PyTorch implementation[19].

### A.3.3 MODEL TRAINING

**RF** was trained using Intel(R) Xeon(R) Gold 6348 CPU @ 2.60GHz. A grid search was conducted for the hyperparameters "n_estimators" with values of [100, 200, 300] and "max_depth" with options [10, 15, 20]. All other hyperparameters remained as the default settings in scikit-learn v1.3.0.

All neural network-based models, i.e., **MLP**, **LSTM** and **PLM-based models**, were trained with the following settings: Adam Kingma & Ba (2014) optimizer without weight decay, the cross-entropy loss function, and employing the early-stop strategy with a patience of five epochs during the training process. The epoch showcasing the best performance on the validation set was chosen for testing.

**MLP** and **LSTM** models were trained on a single NVIDIA RTX 3090 GPU. The batch size was set to 1024 for MLP and 128 for LSTM. The learning rate for both models was selected from the set [8e-3, 5e-3, 3e-3, 1e-3], based on preliminary tuning experiments that prioritized convergence stability and performance.

**PLM-based models** were trained across four NVIDIA RTX 3090 GPUs. To ensure a balance between training efficiency and model accuracy, the batch size was fixed at 32 for levels I-IV on each GPU. For the ESM2-150M model, the batch size was reduced to 8 at level IV per GPU to avoid memory overflow. Learning rates for these models were chosen from [1e-4, 8e-5, 5e-5, 3e-5, 1e-5], following an extensive grid search aimed at optimizing both convergence speed and final model performance.

---

[16]https://huggingface.co/facebook/esm2_t6_8M_UR50D
[17]https://huggingface.co/facebook/esm2_t12_35M_UR50D
[18]https://huggingface.co/facebook/esm2_t30_150M_UR50D
[19]https://github.com/songlab-cal/tape

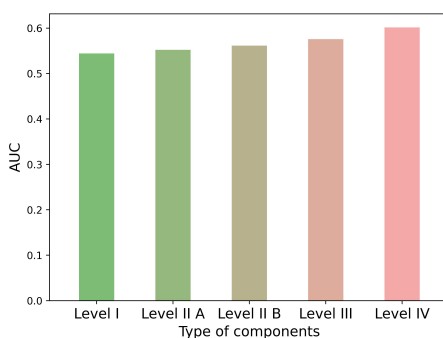 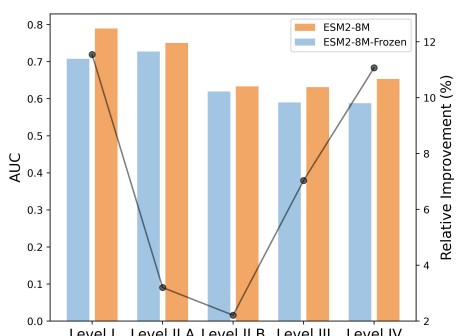

Figure 8: Performance of ESM2-35M on the Hi-TPH level IV dataset using components from different levels.

Figure 9: Performance of ESM2-8M on Hi-TPH datasets. Two variants are evaluated: one with the PLM's parameters frozen (ESM2-8M-Frozen) and the other fine-tuned (ESM2-8M).

## A.4 ADDITIONAL BENCHMARK RESULTS

### A.4.1 CONTRIBUTION OF ADDITIONAL COMPONENTS

To further investigate the contribution of additional input features on TCR-pHLA binding prediction, we performed analogous experiments to those conducted with the ESM2-35M model, but this time utilizing the MLP and ESM2-8M models. We compared the results obtained by using only the peptide and CDR3 $\beta$ sequences versus incorporating all available input components, as shown in Fig. 6 and Fig. 7. Consistent with the findings for ESM2-35M, we observed the following:

- For the level II A task, the inclusion of the HLA sequence contributed little to the model performance, and in some cases even yielded a slightly negative effect.

- In contrast, for the more complex level II B, III, and IV tasks, incorporating the additional input components, such as CDR3 $\alpha$ and other available features, led to a significant improvement in the predictive performance of the models.

These results further corroborate the conclusions drawn from the main analysis using the ESM2-35M model, highlighting the differential importance of incorporating various input features for TCR-pHLA binding prediction across the different task difficulty levels.

Additionally, we validated the effect of progressively incorporating additional components of different levels on ESM2-35M using the level IV dataset. We first filter the level IV data and remove duplicates, focusing solely on peptide and CDR3 $\beta$. With 22,099 TCR-pHLA pairs remaining, components of different levels are used as model inputs. As illustrated in Fig. 8, the constant introduction of additional components results in a consistent enhancement of the model performance.

### A.4.2 IMPACT OF FINE-TUNING PLMS

We also explore the impact of fine-tuning the ESM2-8M model. Fig. 9 shows that fine-tuning ESM2-8M enhances the ability to utilise the TCR-pHLA information at different levels, especially at level I and IV. Moreover, comparing ESM2-8M to ESM2-35M reveals that fine-tuning provides more relative improvement for the model with more trainable parameters.

## A.5 PERFORMANCE OF THE ON-THE-FLY MISPAIRING METHOD

By generating negative samples during the algorithm's execution instead of relying on a predetermined set, the on-the-fly mispairing method is less prone to overfitting to specific negative examples. Additionally, the flexibility to adjust the size of negative samples as training progresses allows for a better balance between positive and negative classes, which is essential for robust model performance. To further validate the advantages, we conducted experiments using the ESM2-35M model

| | Level I | | | Level II A | | | Level II B | | | Level III | | | Level IV | | |
|---|---|---|---|---|---|---|---|---|---|---|---|---|---|---|---|
| | OM | FN | RI (%) | OM | FN | RI (%) | OM | FN | RI (%) | OM | FN | RI (%) | OM | FN | RI (%) |
| AUC | 0.9439 | 0.9237 | **2.18** | 0.7548 | 0.7528 | **0.27** | 0.6404 | 0.6318 | **1.37** | 0.6363 | 0.6221 | **2.29** | 0.6510 | 0.6381 | **2.01** |
| ACC | 0.8884 | 0.8577 | **3.59** | 0.6794 | 0.6752 | **0.62** | 0.5887 | 0.5830 | **0.97** | 0.5913 | 0.5757 | **2.71** | 0.6044 | 0.5860 | **3.14** |
| MCC | 0.7776 | 0.7183 | **8.27** | 0.3666 | 0.3503 | **4.65** | 0.1775 | 0.1661 | **6.89** | 0.1827 | 0.1515 | **20.59** | 0.2174 | 0.1722 | **26.20** |
| F1 | 0.8859 | 0.8638 | **2.55** | 0.7094 | 0.6734 | **5.34** | 0.5810 | 0.5878 | -1.16 | 0.5962 | 0.5695 | **4.69** | 0.6526 | 0.5948 | **9.71** |

Table 5: Performance comparison of ESM2-35M using On-the-Fly Mispairing (OM) and Fixed Negative (FN) approaches. RI represents the relative improvement of OM compared to FN.

to compare the On-the-Fly mispairing (OM) with the fixed negative (FN) methods. Our results demonstrate that the OM approach enhances predictive performance, evidenced by improvements w.r.t. AUC, accuracy (ACC), Matthews correlation coefficient (MCC), and F1 score. The relative improvements achieved by the OM method over FN are consistent across all levels of the dataset, highlighting its robustness and effectiveness. The details are summarized in Table 5.

