# OpenReview forum: "Hi-TPH: A Large-Scale Hierarchical Dataset for TCR-pHLA Binding Prediction"
_ICLR.cc/2025/Conference — Submitted to ICLR 2025_

### Official Review · Reviewer_7FS3 · 2024-10-31

**Soundness:** 3
**Presentation:** 3
**Contribution:** 2
**Rating:** 5
**Confidence:** 4

**Summary:**

This paper introduces Hi-TPH, a large-scale hierarchical dataset for predicting T cell receptor (TCR) and peptide-human leukocyte antigen (pHLA) binding interactions. The dataset is organized across multiple levels, each adding more components (e.g., full TCR α and β chains, HLA sequences) to improve prediction model training. Additionally, the authors propose an on-the-fly mispairing method to generate negative samples dynamically, improving model robustness. The work establishes baselines using various models, providing a valuable resource for future research in immunotherapy applications such as personalized vaccines.

**Strengths:**

1.	Hi-TPH is structured to support a variety of modeling approaches, with hierarchical levels that add predictive components
2.	Extensive evaluation across multiple models provides essential benchmarks and insights into model performance across dataset levels.
3.	The dataset and proposed method could support applications in precision medicine, especially in developing immunotherapies that rely on specific TCR-pHLA interactions.

**Weaknesses:**

The rationale for including specific components at different levels, such as why certain levels exclude the HLA component, could be elaborated to clarify the dataset structure.

**Questions:**

1.	Could you clarify the specific innovations compared to the datasets used in pMTnet and TransPHLA-AOMP?
2.	Could the authors clarify the motivation behind the hierarchical structuring of the Hi-TPH dataset? Specifically, why were certain components prioritized in each level?
3.	How does the inclusion of full TCR α and β chain sequences at higher levels contribute to model performance compared to focusing solely on the CDR3 regions?

---

### Official Review · Reviewer_zUEk · 2024-11-01

**Soundness:** 2
**Presentation:** 3
**Contribution:** 2
**Rating:** 5
**Confidence:** 4

**Summary:**

This paper presents a dataset for of TCR-pMHC binding pairs and benchmarks several. The dataset has a hierarchical structure depending on how many components are available. A handful of models are benchmarked against this dataset, which illustrate the benefit of including information beyond just the CDR3$\beta$ and peptide, and fine-tuning PLMs.

**Strengths:**

Hi-TPH contains significantly more TCRs, pMHCs, and pairs than datasets used for training in other papers. The composition of the hierarchical datasets is clear, and the inclusion of full sequences may be helpful for research in TCR-pMHC binding models. There is a good discussion of the impact of including different sequence information for datasets lower down the hierarchy, and of the impact of fine-tuning PLMs. The filtering of the positive dataset is clear.

**Weaknesses:**

Many of the papers listed in Table 1 claim to pull from exactly the same, or a very similar, set of original datasets as Hi-TPH (VDJdb, IEDB, McPAS, MIRA). The reader might therefore expect a similar number of samples in the datasets used in these papers as in Hi-TPH. I suspect it may have to do with the inclusion of murine data in Hi-TPH, but the paper could be improved by including a thorough discussion of this discrepancy. The introduction of this dataset would be more impactful if the authors could show that including this additional data improves performance at the binding prediction task.

The dataset is "randomly" split into training, validation, and test sets, but it is not clear the steps taken to prevent data leakage: can the same CDR3$\beta$/peptide/etc. occur in the same splits? This is a non-trivial task for the datasets lower in the hierarchy, and could do with more discussion. Leakage of sequences lower down in the hierarchy that cannot exist higher up the hierarchy may impact the benchmarking results. Moreover, there is a discussion of unseen peptides in the test set, but it is unclear how these unseen peptides are chosen: are they just from peptides which have by chance not appeared in the training dataset? If so, they are more likely to belong to the long tail described in Section 3.2, which could introduce a bias, and the unseen benchmark might improve by selecting the peptides in a more appropriate way.

Although the "on-the-fly mispairing" is used in training, it is not clear how the negatives in the test set are constructed - is the test set deterministic?

This paper lists several models in Table 1, but these are not included in the benchmark.

**Questions:**

1. Why are the datasets in Table 1 significantly smaller than Hi-TPH?
2. How is data leakage prevented in the dataset splits?
3. How was the unseen benchmark constructed? How were the negatives in the test set constructed?
4. How do the other models in Table 1 perform on this benchmark?

---

### Official Review · Reviewer_c4Ep · 2024-11-04

**Soundness:** 3
**Presentation:** 3
**Contribution:** 1
**Rating:** 3
**Confidence:** 5

**Summary:**

This manuscript is a dataset paper that proposes to make available an experimental dataset that can be used in the machine learning community to develop new tools to predict interactions between T cell receptors and peptide-MHC molecules. This particular interaction is of central importance in order to understand how T-cell mediated immune systems recognises viruses, pathogens, cancer cells, etc. This paper also develops and applies some baseline models to evaluate the relative performance of different types of machine learning models on this task.

**Strengths:**

Recognition of virus or pathogen derived peptides by the immune system is important and can e.g. provide diagnostic tools or tools to design tailored treatments. The T cell centered immunity does that recognition using T cell receptors (TCR) that can recognise peptides that are presented on the surface of e.g. dendritic cells by the major histocompatibility complex (MHC). The TCR, peptide and MHC are highly variable which makes it difficult to understand and predict these interactions. There is only little structural data available from this triplet, but fortunately experimental data has been collected that characterise these interactions either as binary response variable or continuous binding affinity measurement (not available in large quantities and not included in this work). The field has two main databases that has collected this information, VDJdb and IEDB, as well as a few other smaller datasets. This manuscript contributes by making this dataset more easily accessible for machine learning researchers who may not be familiar with immunology or otherwise may feel difficult to access the data from the database directly.

The TCR, peptide ,MHC interaction prediction has attracted lots of attention recently in the bioinformatics/ML community. In terms of originality and significance, the impact of this work is weak though because essentially all previous methods have used exactly the same databases. Depending on the pre-processing filters used by earlier studies, or depending on how much data has been in these databases at the time of publizing the earlier studies, determines “how big datasets” the previous studies have used. Claiming large improvements in terms of dataset sizes are therefore subjective, and anyways some recent papers that used almost as many data points as this manuscript are not cited here.

The quality and clarity of data collection looks good.

**Weaknesses:**

The number of papers published on this topic has increased during the recent years.  Authors try to be extensive in describing some of them but also ignore several recent contributions.  Authors note for example (line 70), that majority of previous works have focused on using only the complementarity determining region 3 (CDR3). There are several earlier works, some of which are also cited in this manuscript, that try to utilize all parts of the TCR alpha and beta chains, such as ERGO-II (cited here), Titan, TCRGP, epi-TRACE, DEEPTCR, and perhaps some others.

Manuscript is a bit unclear of whether it is primarily the CRD3 that is the key determinant of the interaction, or do other TCR parts also contribute to the interactions. Glanville et al. (cited here) carried out a structural analysis of contacts between amino acids between different parts of the TCR, peptide MHC, using crystal structures, suggesting that also parts beyond CDR3 are important, and earlier studies have observed similar in terms of features important in their ML methods (e.g. ERGO-II and epi-TRACE, perhaps others that I do not remember now).

As mentioned above, the number of TCR, peptide-MHC pairs analysed in previous studies has been affected by pre-processing steps that the authors have decided to use, as well as the number of data points in the VDJdb and IEDB databases at the time of earlier studies. Claiming significant increase in the number of data points is somehow subjective, because essentially all previous papers have been using the exact same datasets. If one downloads the VDJdb and IEDB datasets, one immediately gets about 200k data points (this is an estimate, I didn’t do check the numbers now), or about 20k datapoints that would be in level 4 (using the terminology of this manuscript), which are comparable numbers reported here. Previous papers that have released their code have also made data processing scripts available, so earlier scripts makes such datasets also automatically available. I understand that this manuscript tries to make data even more easliy accessible to ML community, but I am not sure the contribution of this manuscript is significant enough to be a separate publication, as many earlier works have done exactly the same task.

One challenge in this ML prediction problem is that the experimental dataset contains only positive data points and negatives are typically artificially generated. Authors propose here to use so-called on-the-fly mispairing for that purpose. In practise, that means doing the mispairing randomly e.g. for each training epoch. Some of the previous methods may do that as well, e.g. if I remember correctly ImReg tool only takes the positive data points as an input. Whether it resamples the negative for each epoch, I would need to check from the code.

Some of the earlier methods, can also utilize partial information about TCRs (i.e., when the full length protein is not available for each data point), e.g. a recent method called TULIP.

Long-tail distribution of binding TCRs for different peptides is a known challenge in these datasets, but authors do not provide any solutions.

Authors conclude that HLA information (that encodes MHC genes) may not provide additional predictive capabilities. This may be too strong of statement, and perhaps rather reflect limitations of the current datasets, where peptide is typically measured only in a single HLA context, resulting in little variability for ML models to learn from.

Results. Authors provide baseline comparison results for the prediction task using different ML methods, including random forest, MLP, and language-based methods. It seems that apart from TAPE-BERT, none of the baseline methods used exactly correspond to any of the published methods. The comparisons would make more sense and would be more valuable it they were carried using published methods that are all tailored for this task.

**Questions:**

-

---

### Official Review · Reviewer_r3kP · 2024-11-09

**Soundness:** 3
**Presentation:** 3
**Contribution:** 3
**Rating:** 5
**Confidence:** 3

**Summary:**

This paper introduces a comprehensive dataset designed to enhance the prediction of T cell receptor (TCR) and peptide-human leukocyte antigen (pHLA) interactions, which are crucial for T cell-mediated immunity. The Hi-TPH dataset addresses the limitations of previous models by incorporating a broader range of protein sequences and gene annotations, providing a more detailed understanding of the TCR-pHLA binding process.

**Strengths:**

The new dataset seems to be helpful in understanding the problem. Hi-TPH is a large-scale dataset that provides a comprehensive view of TCR-pHLA interactions by including multiple levels of protein sequences and gene annotations. The dataset's stratification into different levels allows for a nuanced analysis of the binding process, enabling researchers to understand the impact of various components on binding affinity. The benchmark results are helpful, which establishes valuable baselines for future research and development in TCR-pHLA binding prediction.

**Weaknesses:**

The dataset lacks continuous confidence scores that differentiate between strong and weak bindings, which may limit the models' ability to capture the full spectrum of binding affinities.  The benchmark may have not been well tuned. Why the larger ESM model performs worse than small models? While the dataset is large, the ability of models to generalize to new peptides remains a challenge, indicating that the current dataset may not fully capture the diversity of TCR-pHLA interactions.

**Questions:**

Q1: Why the larger ESM model performs worse than small models?

Q2: Considering the limited number of peptide, how can the model generalize to new peptides?

Q3: Randomly spliting dataset into 8:1:1 training, validation, and test sets may lead to data leakage. Why not use sequence or structure simiarity to partition the data?

Q4: Will you be maintaining the dataset? What is the plan?

---

### Meta-Review · Area_Chair_kyYy · 2024-12-10

**Metareview:**

This paper contributes a large dataset for predicting TCR-pHLA binding, an important problem in computational biology which can benefit from Machine Learning. The reviewers while appreciative raised several concerns. The concerns were not addressed;the author(s) did not submit any rebuttal.
There is consensus that the current manuscript maynot be suitable for ICLR.

**Additional Comments On Reviewer Discussion:**

The author(s) did not submit a rebuttal and  the concerns of the referees were not addressed.

---

### Decision · Program_Chairs · 2025-01-22

Reject